# Genetic Relationships and Reproductive Traits of Romanian Populations of Silver Fir (*Abies alba*): Implications for the Sustainable Management of Local Populations

**Irina Maria Todea Morar** [1]**, Stephanie Rensen** [2]**, Santiago Vilanova** [2]**, Monica Boscaiu** [3]**, Liviu Holonec** [1]**, Adriana F. Sestras** [1]**, Oscar Vicente** [2]**, Jaime Prohens** [2]**, Radu E. Sestras** [1,*]**and Mariola Plazas** [2,*]

[1] Faculty of Horticulture, University of Agricultural Sciences and Veterinary Medicine of Cluj-Napoca, 400372 Cluj-Napoca, Romania; irina.todea@usamvcluj.ro (I.M.T.M.); lholonec@usamvcluj.ro (L.H.); adriana.sestras@usamvcluj.ro (A.F.S.)

[2] Institute for the Conservation and Improvement of Valencian Agrodiversity (COMAV), Universitat Politècnica de València, Camino de Vera s/n, 46022 Valencia, Spain; stephanierensen@hotmail.com (S.R.); sanvina@upvnet.upv.es (S.V.); ovicente@upvnet.upv.es (O.V.); jprohens@btc.upv.es (J.P.)

[3] Mediterranean Agroforestry Institute (IAM), Universitat Politècnica de València, Camino de Vera s/n, 46022 Valencia, Spain; mobosnea@eaf.upv.es

* Correspondence: rsestras@usamvcluj.ro (R.E.S.); maplaav@btc.upv.es (M.P.)

**Abstract:** Assessment of the diversity of reproductive traits and genetic variation is of great relevance to the conservation of genetic resources and management of silver fir (*Abies alba*) populations. We have evaluated reproductive characteristics associated with female cones and seed morphology, as well as seed germination after subjecting seeds to five storage methods, in nine Romanian populations of *A. alba*. The genetic diversity of the populations was assessed with 12 polymorphic simple sequence repeat (SSR) markers. We detected significant differences between populations for all reproductive traits and considerable differences in seed germination and storage methods; seed storage in wet sand was the method resulting in the highest germination in all populations. Genomic SSRs (gSSRs) were more informative on average than expressed sequence tag SSRs (EST-SSRs) in the populations studied. The nine populations were genetically diverse, with an average number of alleles (N) per SSR locus between 3.50 and 4.83. The observed heterozygosity (Ho) in the nine populations was always lower than the expected heterozygosity (He), which resulted in values of the inbreeding coefficient (Fis) between 0.261 and 0.709. Genetic distances between populations ranged between 0.077 and 0.410. The cluster analysis based on genetic distances did not group accessions according to their geographical proximity, and despite a positive trend, the correlation between geographic and genetic distances was non-significant. The results of an analysis of molecular variance (AMOVA) revealed that only 9.1% of the total molecular variance is attributable to differences between populations. This low degree of genetic differentiation between populations is confirmed by the intermingling of individuals of different populations in a principal coordinate analysis (PCoA). We found evidence of a positive relationship between He and germination, as well as a negative one between Fis and germination, suggesting that populations with low diversity and high consanguinity may have a reduced fitness and long-term viability. The results are relevant for the conservation and management of local genetic resources and populations, as well as for reforestation programmes of silver fir.

**Keywords:** silver fir; *Abies alba*; germination; SSRs; genetic diversity; inbreeding

## 1. Introduction

In Europe, the genus *Abies* Mill. has a circum-Mediterranean distribution [1]. Among *Abies* species, silver fir (*Abies alba* Mill.) is the only one with a broad range of distribution [1,2]. Silver fir is one of the tallest tree species of the *Abies* genus, growing up to 60 m under favorable conditions [3], and is usually found at altitudes between 500 and 2000 m, requiring relatively high moisture conditions all year round. Compared to Mediterranean *Abies* species, silver fir generally prefers cooler and moister conditions [4]. Apart from its economic value, it is considered an important species in many mountain forests. In this respect, silver fir contributes to high biodiversity in forest ecosystems because of its ability to coexist with many other tree species in ecologically rich forests [5].

Silver fir ecotypes display high variation for some stress traits [5], but silver fir does not seem to be highly tolerant to water deficit or salinity in the seedling stage [6,7]. The current scenario of climate change may affect the viability of *A. alba* populations and, therefore, improvements in forest management are needed for effective conservation of this species [5]. Silver fir usually coexists with other tree species, particularly with Norway spruce (*Picea abies* (L.)) and Karst and European beech (*Fagus sylvatica* L.), although on occasion it can be found in pure stands [8]. Silver fir populations have been declining over the last 200 years [9], and measures for the conservation and sustainable management of its populations are required. If in the remote past the area occupied by fir probably represented 10–15% of the forest area of Romania, in 1929 its share was reduced to 6.5%, and in 1984 the tree occupied 5.12% of Romanian forests. In 1989, it occupied about 317 thousand hectares, i.e., 5.1%, and in 2003 it occupied 307 thousand hectares, which means 5.0% of the area covered by forests [10]. Even if this proportion is relatively low, this species contributes with other forest species to ensure the multiple functions of a forest: production, landscape, ecological and environmental protection, cultural, educational and recreational [11,12]. All these functions can be components of the concept of sustainable and multifunctional forest proposed for Central Europe [13].

Production of high quality seeds is of great relevance to the maintenance of viable populations of forest trees, as well as reforestation purposes. Seed production of forest trees is generally influenced by many internal and external elements, and the germination and regeneration of silver fir in different ecological conditions depend on various factors, such as temperature, light and dissemination patterns [14]. In the case of silver fir, after entering the reproductive age under natural conditions, there is a peak of seed production every 4–5 years, which can be reduced to 2–3 years in plantations [15]. Previous studies have revealed substantial differences in the morphological characteristics of silver fir seeds and female cones, as well as in the germination capacity of seeds from different populations [16–19].

Evaluation of genetic diversity is of high relevance for the management of *A. alba* genetic resources. Several surveys of the genetic diversity and structure of silver fir populations have been performed using DNA markers [20–26]. These studies have provided relevant information, revealing that *A. alba* populations generally display high genetic diversity and low genetic differentiation among populations from the same geographical region.

In the present study, we evaluated the characteristics of the structures (female cones) where the primary reproductive material (seeds) is contained, seed germination using different storage systems, and assessed genetic diversity and genetic structure with simple sequence repeat (SSR) markers in nine populations of silver fir from Romania. Based on both datasets, we have evaluated the relationship between germination, an important parameter associated with fitness and population viability [27], and genetic parameters related to population diversity and inbreeding [28]. The results will be of interest when it comes to devising strategies for the conservation and management of genetic resources, reforestation and sustainable exploitation of silver fir in Romania.

## 2. Materials and Methods

### 2.1. Plant Material

Nine native populations from different geographical areas, several corresponding to the Romanian gene reserve forests and seed stands included in National Catalogues of Forest Genetic Resources and Forest Reproductive Materials [29], were used in the present study (Table 1, Figure 1). All populations were collected in the Carpathian mountain range areas. Within Romania, five of the populations (1, 3, 6, 7, and 8) are situated in the northwestern part, one (5) in the northern region, and three (2, 4 and 9) in the central part (Figure 1). Different ranges of distances among populations were chosen to evaluate the effect of geographical distance on genetic distance.

**Table 1.** Origin and geographical coordinates of nine silver fir (*Abies alba*) populations native to Romania, according to the National Catalogue of Forest Reproductive Materials, Bucharest [29], with the exception of populations 7 and 8.

| Code | Population | County | Administrative Location | Latitude/Longitude | Altitude (m asl) |
|------|-----------|--------|-------------------------|--------------------|------------------|
| 1 | Valea Bistrei | Alba | OSP. Abrud, UP III, u.a. 228B | 46°27′ N/23°01′ E | 1050–1325 |
| 2 | Iod | Mureș | OS. Răstolnița UP VI-67 A | 46°56′ N/24°59′ E | 670–1063 |
| 3 | Someșul Rece | Cluj | OS. Someșul Rece UP I u.a. 92A | 46°38′ N/23°14′ E | 690–1250 |
| 4 | Avrig | Sibiu | OS Izvorul Florii UP III u.a 75A | 45°37′ N/24°27′ E | 900–1150 |
| 5 | Budescu | Maramureș | OS. Poieni, UP IV, u.a 96 A | 47°54′ N/24°36′ E | 860–1120 |
| 6 | Sohodol | Alba | OSP. Abrud UP 18 C | 46°20′ N/23°06′ E | 870–1030 |
| 7 | Valea Morii | Alba | OSP. Abrud U V, u.a 39 | 46°19′ N/22°56′ E | 1080–982 |
| 8 | Gârda Seacă | Alba | OS. Gârda UP VI, u.a 20 H | 46°31′ N/22°46′ E | 1090–1285 |
| 9 | Sadu | Sibiu | OS. Valea Sadului UP II, u.a 75 A | 45°35′ N/23°55′ E | 1100–1350 |

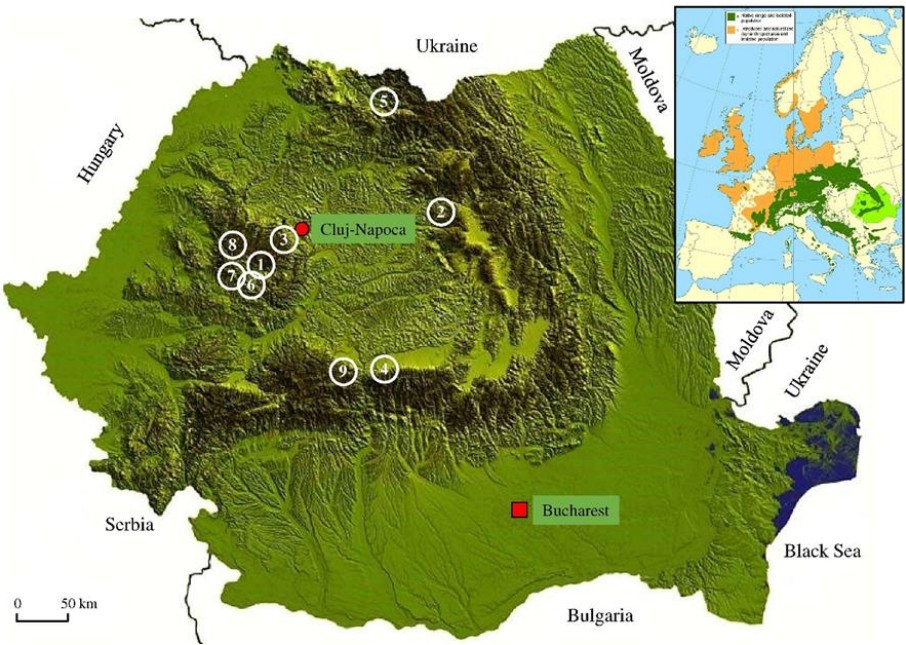

**Figure 1.** Map displaying the location of origin of the nine populations of silver fir (*Abies alba*) from Romania. In the corner icon on the top right, the native distribution of silver fir populations in Europe is represented in green (after Caudullo et al. [30]).

### 2.2. Characterization of Female Cones, Seeds, and Seed Germination

Female cones of *A. alba*, collected in bulk from healthy adult trees from the nine selected populations, were stored for a few days in the laboratory at a constant temperature of 20 °C, before the seeds were extracted individually from each of them. A total of 36 cones from each population, randomly taken from the bulk sampling, were used for the measurements. For each female cone, the following traits were determined: cone length (mm), cone diameter (mm), cone length/diameter ratio, number of seeds per cone, seed length (mm), and seed weight (mg). Measures of seed length and weight included the wing. All traits were expressed as mean values.

The germination capacity of the seeds from the nine silver fir populations was assessed after subjecting the seeds to five storage methods for three months during the dormancy period: a) the standard INCDS (National Institute for Research and Development in Forestry) method [15], with low humidity and a constant temperature of 0 °C; b) dry sand layer in a sheltered place at room temperature [31]; c) control (sand and soil) with a layer of soil and a layer of sand in a sheltered place at a temperature of 2–5 °C; d) in wet sand at a temperature of 2–5 °C [32]; and, e) open spaces in cold rooms with a maximum relative humidity of 13% and a variable thermal regime [33]. For evaluation of germination, the *A. alba* seeds were watered every second day with tap water at a rate of 1 L/tray and held under controlled conditions at a temperature of 20 °C. Four replicates, with 50 seeds per replicate, were used for each combination of population and storage method.

### 2.3. DNA Extraction and SSR Genotyping

Total genomic DNA was isolated from fresh young needles using a modification of the Cetyl Trimethyl Ammonium Bromide (CTAB) extraction method previously described [34]. Briefly, homogenisation, lysis and protein precipitation steps were performed as described by Doyle and Doyle [35]. The washing and precipitation steps were carried out using a homemade silica matrix [34]. The DNA concentration and quality parameters (230/260 and 260/280 nm absorbance ratios) were measured with a NanoDrop ND-1000 spectrophotometer (Thermo Fisher Scientific, Waltham, MA, USA). DNA quality was further evaluated in a 1% agarose gel. The extracted DNA was diluted to a concentration of 50 ng/μL and stored at −20 °C until used.

Twenty-four SSR markers were considered for the genotyping of the samples (n = 5 for populations 1, 3, 8, and 9; n = 4 for populations 2, 4, 5, 6, and 7). Sixteen SSRs corresponded to previously developed expressed sequence tag SSRs (EST-SSRs) [21] and eight to genomic SSRs (gSSRs) described by different authors [21,36,37]. SSRs were organized in three multiplexes using the same layout as in [21], with EST-SSRs distributed in multiplexes A and B and gSSRs in multiplex C (Table 2). The PCR products obtained were diluted based on the amplification signal and then sequenced with an ABI 3130XL genetic analyser (Applied Biosystems, Foster City, CA, USA). The size calling of the electropherograms was performed using GeneScan® software (Thermo Fisher Scientific, Version 2.0 CA, USA). Then, polymorphisms were examined with Genotyper® DNA Fragment Analysis software (Thermo Fisher Scientific, Version 2.0 CA, USA).

### 2.4. Data Analysis

For the characterisation of data relating to female cones and seeds, a one-way ANOVA was performed to assess differences between populations. For germination data, a two-way ANOVA, with population and seed storage method as the main factors, was performed. Significance of differences between the main factors from the ANOVA analyses was evaluated with a Student–Newman–Keuls multiple range test ($p < 0.05$).

**Table 2.** Simple sequence repeat (SSR) markers used for the evaluation of diversity in nine Romanian populations of silver fir (*Abies alba*) and their characteristics [21].

| Locus | Primer Sequence (5′–3′) | Dye [a] | Size Range (bp) |
|---|---|---|---|
| Multiplex A (EST-SSRs) | | | |
| Aat01 | F: CCATGTCTCCGATTTCCAGT<br>R: GGCCTAACGAAAGCAGAATC | FAM | 103–127 |
| Aat02 | F: AGAAGATTTCCCGGCTTTTC<br>R: ATCCAGACAGCGAACTTTGG | VIC | 123–129 |
| Aat03 | F: TCCCCATGGTTTGGTTAAAA<br>R: CGAAGAAAATGTTGCGGAAT | PET | 149–161 |
| Aat04 | F: CCATGTATGGTGCTCCTCCT<br>R: CCTTCATTGCAGAAAAGCAA | FAM | 158–191 |
| Aat05 | F: AGCATCCACATTCCGTAACC<br>R: AGTTGACCGTTGGAGAGCAG | VIC | 177–192 |
| Aat06 | F: TTATGCGGAGCAGTTCTGTG<br>R: TGTTGCTGGCGTACTGGTAG | NED | 196–214 |
| Aat07 | F: GCTAGCAGAACCCTGGAATG<br>R: GGTGGGATATTTCCAGCAAG | PET | 219–241 |
| Aat08 | F: ACTCCATCACGGTGGTCTTC<br>R: GCCATTCAGGCTCTCAGTTC | NED | 302–312 |
| Multiplex B (EST-SSRs) | | | |
| Aat09 | F: CAGATCCTCCCACATCCAAC<br>R: TGACACCACAGGAAACCATC | NED | 150–156 |
| Aat10 | F: GAGCACGATGAAGAGGAAGC<br>R: AAAACCCCCACGCGGTAT | FAM | 226–250 |
| Aat11 | F: AGCGTTGATTGGAAGCAGTC<br>R: GAAGCATGGTGTCGTTGTTG | VIC | 255–270 |
| Aat12 | F: ATCCATATCTCCTGCCTTGC<br>R: CTTTCCAGGTGATCTGATTGC | PET | 303–349 |
| Aat13 | F: ACTCAAAGCCAAGCTGGAGA<br>R: TGCATAAGACAGCCGAGTCA | FAM | 326–342 |
| Aat14 | F: GACTGGGGATCCTGCTGTTA<br>R: AGAGGAGGCAGCCCATACAT | VIC | 358-394 |
| Aat15 | F: AGGAGGAGGTTCAGCATGTC<br>R: CTTGCTCTCTGACCCAGTTG | NED | 361–373 |
| Aat16 | F: AACCACCGCTGATATTTTGG<br>R: GGGTTCAAGAAATGGGAATG | PET | 427–430 |
| Multiplex C (gSSRs) | | | |
| SFg6 | F: GTAACAATAAAAGGAAGCTACG<br>R: TGTGACACATTGGACACC | VIC | 103–111 |
| SF324 | F: TTTGAACGGAAATCAAATTCC<br>R: AAGAACGACACCATTCTCAC | PET | 105–120 |
| NFF7 | F: CCCAAACTGGAAGATTGGAC<br>R: ATCGCCATCCATCATCAGA | VIC | 116–174 |
| SFb5 | F: AAAAAGCATCACTTTTCTCG<br>R: AAGAGGAGGGGAGTTACAAG | FAM | 138–160 |
| SFb4 | F: GCCTTTGCAACATAATTGG<br>R: TCACAATTGTTATGTGTGTGG | NED | 149–205 |
| Aag01 | F: GCTTATTCTCACTGCTCGCC<br>R: ATGACTTGAAGGTGGATGCC | PET | 193–250 |
| SF1 | F: TTGACGTGATTAACAATCCA<br>R: AAGAACGACACCATTCTCAC | VIC | 208–229 |
| Aag02 | F: TATTCCTCCACTTGGGTGCT<br>R: GGTGGAGATCCGTATGCAAT | FAM | 208–250 |

[a] FAM = Blue colour; VIC = Green colour; PET = Red colour; NED = Yellow colour.

Regarding the SSR data, the software packages Gene Scan Analysis ver. 2.0 and Genotyper ver 2.0 (Applied Biosystems, Foster City, CA, USA) were used for the identification of alleles sizes. The molecular analysis of SSRs was performed using GenAlEx ver. 6.5 software [38]. The number of alleles (N), number of effective alleles (Ne), frequency of the major allele (f), observed heterozygosity (Ho), expected heterozygosity (He), inbreeding coefficient (Fis), and the polymorphic information content (PIC) were calculated for each SSR marker. As well as this, the percentage of polymorphic loci (P%), average number of alleles per locus (N), number of effective alleles (Ne), observed heterozygosity (Ho), expected heterozygosity (He), and inbreeding coefficient (Fis) were calculated for the nine populations. A phenogram was constructed with R packages poppr, adegenet and phangorn [39–41] using Nei's genetic distance [42] and the neighbour-joining method [43]. Support of the phenogram branches was evaluated using bootstrap analysis with 1000 replications. Analysis of molecular variance (AMOVA [44]) was performed using GenAlEx ver. 6.5 software, and the total molecular variance was partitioned between the following sources of molecular variance: between populations, within populations, and within individuals (caused by heterozygosity at individual loci). Genetic distances between the nine populations were calculated using GenAlEx ver. 6.5 software. Linear regression was used to evaluate the relationship between the genetic and geographical distances of the populations. A principal coordinate analysis (PCoA) was performed to visualise the genetic relationships among individuals using GenAlEx ver. 6.5 software.

To assess the potential relationships between the fitness of the populations and their physical properties, genetic diversity and structure, pair-wise Pearson correlations were calculated between average germination values. These are associated with fitness [45] on the one hand, and seed weight (associated with enhanced germination [46]), expected heterozygosity (He; related to genetic diversity), and the inbreeding coefficient (Fis; associated with consanguinity) [28] on the other.

## 3. Results

### 3.1. Characterisation of Silver Fir Female Cones

Significant ($p < 0.05$) differences were found between the Romanian populations of silver fir for the six traits related to female cone morphology and seed characteristics (Table 3). For example, differences of up to 1.8-fold were found in cone length between populations 1 (79.5 mm) and 7 (139.8 mm), or in cone diameter between populations 8 (33.5 mm) and 5 (61.0 mm). The differences in female cone shape (length/diameter ratio) were more substantial, with a difference of 2.6-fold between populations 1 (1.86) and 9 (4.92). Regarding the number of seeds per cone, they also displayed a difference of 2.6-fold between populations 6 (111.3) and 8 (288.8). The variation in seed length was lower, with differences of up to 1.5-fold between populations 4 (17.1 mm) and 7 (26.0 mm). Finally, for seed weight, there were differences of up to 2.5-fold between populations 4 (38 mg) and 2 (78 mg).

**Table 3.** Morphological characteristics of the female cone, number of seeds per cone, and seed length and weight in the nine Romanian populations of silver fir. Values shown are means per population of the measured parameters.

| Population | Cone Length (mm) | Cone Diameter (mm) | Cone Length/Diameter | Number of Seeds per Cone | Seed Length (mm) | Seed Weight (mg) |
|---|---|---|---|---|---|---|
| 1 | 79.5 a | 42.9 c | 1.86 a | 144.2 b | 20.7 b | 48 b |
| 2 | 133.6 cd | 40.6 bc | 3.29 ab | 213.5 d | 21.0 b | 78 d |
| 3 | 126.3 cd | 38.1 abc | 3.35 ab | 162.9 bc | 23.8 b | 60 bc |
| 4 | 104.6 b | 37.7 abc | 2.80 a | 173.0 bcd | 17.1 a | 38 a |
| 5 | 120.2 c | 61.0 d | 1.99 a | 185.4 bcd | 22.7 b | 52 bc |
| 6 | 94.8 b | 41.7 bc | 2.29 a | 111.3 a | 20.8 b | 57 bc |
| 7 | 139.8 d | 39.2 abc | 3.62 ab | 166.5 bc | 26.0 c | 58 bc |
| 8 | 123.6 cd | 33.5 a | 3.70 ab | 288.8 e | 22.1 b | 60 bc |
| 9 | 134.9 cd | 35.4 ab | 4.92 c | 190.3 cd | 22.3 b | 63 c |

Notes: different letters in each column indicate significant differences, according to the Student–Newman–Keuls multiple range test at $p < 0.05$.

### 3.2. Seed Germination

Significant differences ($p < 0.05$) in the germination percentages were detected in the ANOVA for both main factors—population and seed storage method—as well as for their interaction. The average germination percentages ranged between 19.5% for population 4 and 38.7% for population 8 (Table 4). Populations 1, 2, 3, and 4 had germination percentages below 25%, whereas those of populations 8 and 9 were above 30%. Regarding the seed storage method, the best one was wet sand, which was significantly better than the rest of the methods (42.4% on average); this storage procedure resulted in the highest germination percentages for seeds of all populations, reaching up to 62.5% in population 8. Wet sand was followed by the control (sand and soil), with 26.0% germination; the other three methods had average germination percentages below 23%, with the dry sand layer method (18.6%) being significantly lower than that of the Standard INCDS (22.9%) (Table 4). Although the interaction between both factors (storage method and population) was significant for all populations, wet sand was the best storage method. For the rest of the seed storage methods, cross-over interactions were detected (Table 4).

**Table 4.** Germination data (%) of seeds from nine Romania populations of silver fir, using five seed storage methods. Data are based on four replicates of 50 seeds per combination of population and germination method.

| Population | INCDS Standard | Dry Sand Layer | Control (Sand and Soil) | Wet Sand | Open Spaces in Cold Rooms | Average |
|---|---|---|---|---|---|---|
| 1 | 21.0 | 16.5 | 23.0 | 31.0 | 17.0 | 21.7 ab |
| 2 | 18.5 | 15.5 | 22.5 | 39.5 | 21.0 | 23.4 abc |
| 3 | 11.0 | 15.5 | 15.5 | 42.0 | 15.5 | 19.9 a |
| 4 | 13.0 | 13.0 | 22.5 | 29.0 | 20.0 | 19.5 a |
| 5 | 29.5 | 21.0 | 22.5 | 41.0 | 20.0 | 26.8 bc |
| 6 | 31.5 | 19.5 | 26.0 | 40.5 | 21.0 | 27.7 bc |
| 7 | 22.5 | 15.0 | 26.0 | 47.5 | 20.5 | 26.3 bc |
| 8 | 36.0 | 28.0 | 40.0 | 62.5 | 27.0 | 38.7 d |
| 9 | 23.5 | 23.5 | 36.0 | 49.0 | 27.0 | 31.8 c |
| Average | 22.9 b | 18.6 a | 26.0 c | 42.4 d | 21.0 ab | |

Notes: Average = means for population averages or germination methods separated by different letters are significantly different according to the Student–Newman–Keuls multiple range test at $p <0.05$.

### 3.3. SSR Characterisation

Out of the 24 SSR loci tested in the nine Romanian silver fir populations, one gSSR (SFg6) was discarded because it produced no or very poor amplification. Another 11 SSR markers were not polymorphic in the individuals analysed and, therefore, were not included in the diversity analyses. The remaining 12 SSRs corresponded to eight EST-SSR markers (five from multiplex A and three from multiplex B), and four to gSSR markers (from multiplex C) (Table 5). The number of alleles (N) averaged 5.17 but varied between 2 for markers Aat03, Aat06, Aat13 and Aat16, and 15 for marker NFF7. On average, the number was higher in gSSRs (mean = 8; range 4 to 15) than in EST-SSRs (mean = 3.75; range 2 to 9). The number of effective alleles (Ne) was much lower than N and ranged between 1.17 for EST-SSR marker Aat13 and 2.86 for gSSR marker NFF7. The frequency of the major allele (f) ranged between 0.278 for gSSR marker NFF7 and 0.900 for EST-SSR marker Aat13 (Table 5). The observed heterozygosity (Ho) ranged between 0.000 (Aat06, Aat13, SFb5, NFF7, and SF324) and 0.426 (Aat03) and had an average value of 0.140, whereas the expected heterozygosity (He) values ranged between 0.095 (Aat13) and 0.628 (NFF7), with an average value of 0.329 (Table 5). For all markers except two (Aat03 and Aat16), Ho values were lower than those of He. As a result, the inbreeding coefficient values (Fis) ranged between −0.298 (Aat16) and 1.000 (Aat06, Aat13, SFb5, NFF7, and SF324). PIC values ranged between 0.1469 (Aat16) and 0.8416 (NFF7) (Table 5).

**Table 5.** Genetic diversity statistics of the 12 SSR markers that were successfully amplified and were polymorphic in the evaluation of 40 individuals of silver fir (*Abies alba*) from nine Romanian populations, including the number of alleles (*N*), number of effective alleles (*Ne*), frequency of the major allele (*f*), observed heterozygosity (*Ho*), expected heterozygosity (*He*), inbreeding coefficient (*Fis*), and the polymorphic information content (*PIC*).

| Locus | N | Ne | f | Ho | He | Fis | PIC |
|---|---|---|---|---|---|---|---|
| Aat01 | 3 | 1.65 | 0.725 | 0.344 | 0.348 | 0.012 | 0.3327 |
| Aat03 | 2 | 1.55 | 0.737 | 0.426 | 0.328 | −0.297 | 0.3125 |
| Aat05 | 4 | 1.59 | 0.775 | 0.172 | 0.298 | 0.423 | 0.3448 |
| Aat06 | 2 | 1.23 | 0.775 | 0.000 | 0.133 | 1.000 | 0.2879 |
| Aat07 | 6 | 1.79 | 0.600 | 0.259 | 0.371 | 0.300 | 0.5471 |
| Aat13 | 2 | 1.17 | 0.900 | 0.000 | 0.095 | 1.000 | 0.1638 |
| Aat14 | 9 | 2.13 | 0.303 | 0.206 | 0.480 | 0.572 | 0.7657 |
| Aat16 | 2 | 1.20 | 0.913 | 0.172 | 0.133 | −0.298 | 0.1469 |
| SFb5 | 7 | 1.99 | 0.632 | 0.000 | 0.477 | 1.000 | 0.4936 |
| NFF7 | 15 | 2.86 | 0.278 | 0.000 | 0.628 | 1.000 | 0.8416 |
| SF324 | 4 | 1.41 | 0.857 | 0.000 | 0.202 | 1.000 | 0.2474 |
| Aag01 | 6 | 1.90 | 0.543 | 0.096 | 0.458 | 0.790 | 0.5884 |
| Mean | 5.17 | 1.71 | 0.670 | 0.140 | 0.329 | 0.542 | 0.4227 |

When considering the nine silver fir populations, the percentage of polymorphic loci (P%) ranged between 50.0% in population 2 and 91.7% in populations 7, 8 and 9. The average number of alleles per locus (N) ranged between 3.50 (population 2), and 4.83 (populations 3 and 8) (Table 6). The number of effective alleles (Ne) per locus ranged between 1.58 (population 2) and 2.50 (populations 8 and 9). The observed heterozygosity (Ho) ranged between 0.060 (populations 1 and 4) and 0.260 (population 7), whereas the expected heterozygosity (He) values were always higher than Ho, with values between 0.209 for population 2 and 0.425 for population 9. As a result, inbreeding coefficient (Fis) values ranged between 0.261 for population 7 and 0.709 for population 1 (Table 6).

**Table 6.** Measures of genetic variation within nine populations of silver fir (*Abies alba*) from Romania assessed with 12 SSR markers, including the number of individuals sampled (n), percentage of polymorphic loci (P%), average number of alleles per locus (N), number of effective alleles (Ne), observed heterozygosity (Ho), expected heterozygosity (He), and inbreeding coefficient (Fis).

| Population | n | P% | N | Ne | Ho | He | Fis |
|---|---|---|---|---|---|---|---|
| 1 | 5 | 58.3 | 4.67 | 1.92 | 0.060 | 0.276 | 0.709 |
| 2 | 4 | 50.0 | 3.50 | 1.58 | 0.090 | 0.209 | 0.508 |
| 3 | 5 | 58.3 | 4.83 | 2.00 | 0.090 | 0.261 | 0.512 |
| 4 | 4 | 75.0 | 3.50 | 1.83 | 0.060 | 0.316 | 0.687 |
| 5 | 4 | 75.0 | 3.67 | 2.00 | 0.150 | 0.376 | 0.600 |
| 6 | 4 | 66.7 | 4.00 | 1.92 | 0.100 | 0.299 | 0.524 |
| 7 | 4 | 91.7 | 3.83 | 2.17 | 0.260 | 0.402 | 0.261 |
| 8 | 5 | 91.7 | 4.83 | 2.50 | 0.220 | 0.398 | 0.376 |
| 9 | 5 | 91.7 | 4.67 | 2.50 | 0.230 | 0.425 | 0.387 |
| Mean | | 73.2 | 4.17 | 2.05 | 0.140 | 0.329 | 0.507 |

### 3.4. Genetic Relationships between Populations

The genetic distances between populations ranged from 0.077 (population 2 vs. population 3) to 0.410 (population 5 vs. population 8) (Table 7). We did not observe a clear pattern in the relationship between genetic distances and the area of origin of the populations, with both high and low values calculated for the five populations from the northwestern part of Romania (Table 7). In this respect, multivariate cluster analysis of the nine populations revealed three main clusters supported by a bootstrap value of 100% (Figure 2). One of them contained only one population from the northwestern part of Romania (population 1). The second cluster contained population 5 from northern Romania, one from the northwestern part (population 3) and one of the three from the central part (population 2). Finally, a third cluster contained the rest of populations (see map in Figure 1). The subclusters did not group populations according to their provenance, except a subcluster that grouped two populations (7 and 9) from northwestern Romania. Despite a positive trend in the relationship between genetic distances and geographic distances, the relationship was non-significant ($p = 0.0967$). (Figure 3).

**Table 7.** Matrix of genetic distances [42] between the nine silver fir Romanian populations evaluated.

|         | Pop. 1 | Pop. 2 | Pop. 3 | Pop. 4 | Pop. 5 | Pop. 6 | Pop. 7 | Pop. 8 |
| ------- | ------ | ------ | ------ | ------ | ------ | ------ | ------ | ------ |
| Pop. 2  | 0.159  |        |        |        |        |        |        |        |
| Pop. 3  | 0.089  | 0.077  |        |        |        |        |        |        |
| Pop. 4  | 0.193  | 0.319  | 0.190  |        |        |        |        |        |
| Pop. 5  | 0.184  | 0.178  | 0.153  | 0.223  |        |        |        |        |
| Pop. 6  | 0.186  | 0.265  | 0.190  | 0.210  | 0.264  |        |        |        |
| Pop. 7  | 0.342  | 0.353  | 0.394  | 0.275  | 0.365  | 0.212  |        |        |
| Pop. 8  | 0.269  | 0.345  | 0.291  | 0.260  | 0.410  | 0.097  | 0.200  |        |
| Pop. 9  | 0.288  | 0.311  | 0.316  | 0.280  | 0.404  | 0.341  | 0.232  | 0.321  |

Note: Pop. = population.

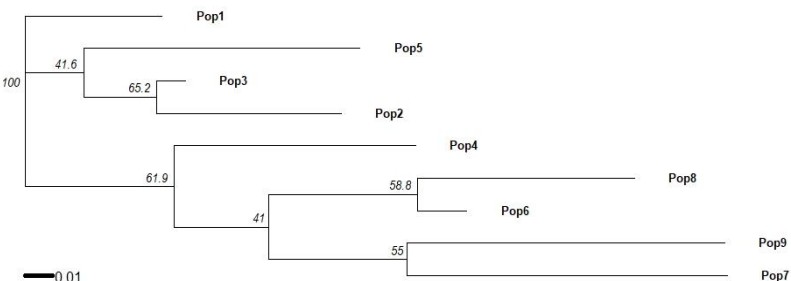

**Figure 2.** Neighbour-joining clustering phenogram of nine populations of silver fir (*Abies alba*) from Romania based on Nei's [42] genetic distances. Bootstrap values (%; 1000 replications) above 50% are indicated at the corresponding nodes.

### 3.5. Genetic Structure

The first and second components of the PCoA account for 12.4% and 11.0% of the total variation, respectively. The PCoA plot shows that individuals from different populations are intermingled, although individuals from populations are not randomly distributed in the PCoA graph (Figure 4). Thus, all individuals from population 3 have positive values for the first coordinate, and all individuals from populations 7 and 9 show negative values for this coordinate. Also, all individuals from population 5 have positive values for the second coordinate, whereas those of population 8 (except for one very low positive value) have negative values for this second component (Figure 4). Partition of the molecular variance in an AMOVA analysis revealed that 9.1% of variance is attributable to differences among

populations and 65.0% to differences within populations (Table 8). The variation within individuals (due to heterozygosity at individual loci) accounts for 25.9% of the total molecular variance.

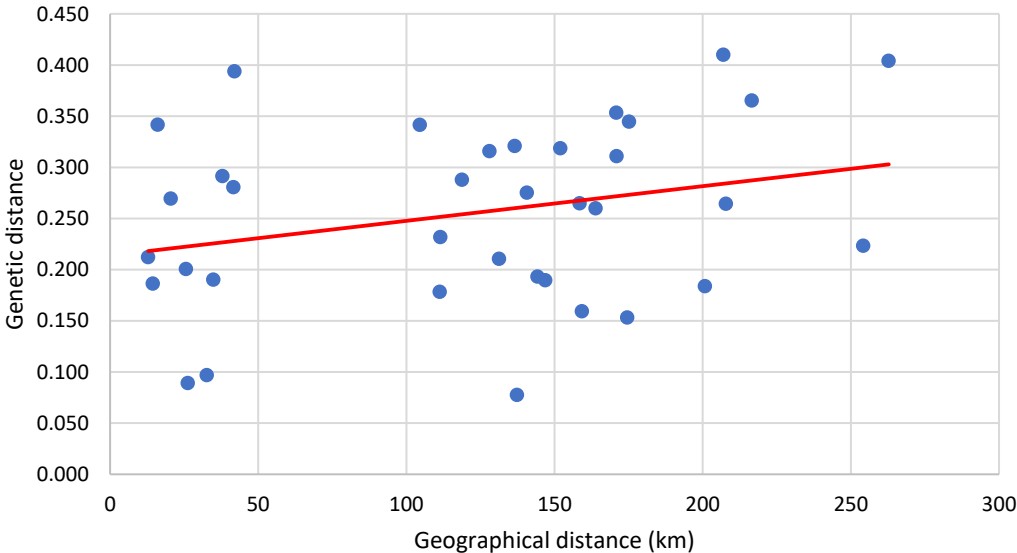

**Figure 3.** Relationship between the geographical distance and Nei's [42] genetic distance estimated with 12 polymorphic SSR markers in the nine populations of silver fir (*Abies alba*) from Romania.

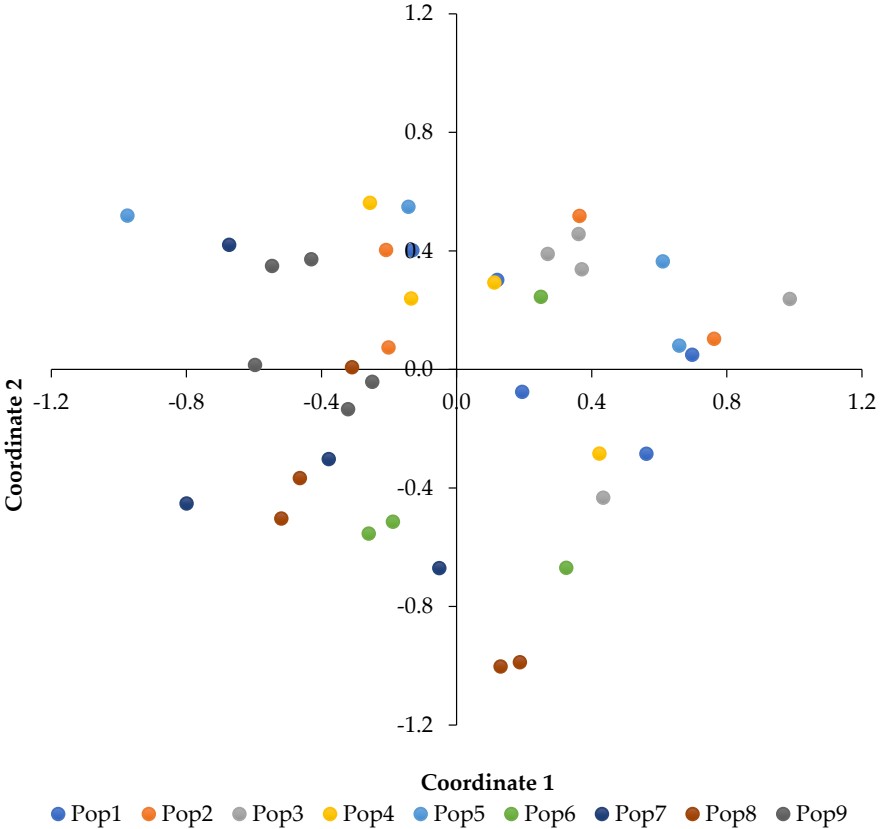

**Figure 4.** Graph displaying the first and second coordinates of a principal coordinate analysis (PCoA) of 40 individuals from nine populations of silver fir (*Abies alba*) from Romania, based on genotyping data for 12 polymorphic SSRs. The percentage of variation accounted for the first and second principal coordinates is 12.4% and 11.0%, respectively.

**Table 8.** Molecular analysis of variance (AMOVA) between populations, between individuals within populations, and within individuals.

| Source of Variation | Degrees of Freedom | Sums of Squares | Mean Squares | Variance of Components | Percentage of Variation |
|---|---|---|---|---|---|
| Total | 79 | 243.575 | | | |
| Between populations | 8 | 59.400 | 7.425 | 0.285 | 9.093 |
| Between individuals | 31 | 151.675 | 4.893 | 2.040 | 65.015 |
| Within individuals | 40 | 32.500 | 0.813 | 0.813 | 25.893 |

*3.6. Relationship between Germination, Seed Weight, and Genetic Parameters*

The Pearson linear correlation coefficient between the average values of germination for the nine populations and their seed weight was non-significant (r = 0.2931; $p$ = 0.4440). However, the correlation between average seed germination and expected heterozygosity (He) was positive and significant (r = 0.6763; $p$ = 0.0455). The correlation between germination and the inbreeding coefficient (Fis) was negative and marginally significant (r = −0.6059; $p$ = 0.0837).

## 4. Discussion

Conservation of certain levels of diversity is important for the maintenance of fitness and viability of populations of forest species [47]. Also, efficient conservation and management of genetic resources of forest species benefit from knowledge of the variation in reproductive traits, seed germination, as well as genetic diversity and structure [48]. In this study, we deal with both issues and try and establish a link between them in Romanian populations of silver fir.

Variation in the morphology of the female cone, number of seeds per cone, and seed size and weight of *A. alba* was described before [49] in populations from the Middle Sudetes. The results obtained in the present study are within the ranges of variation reported in that publication and reveal that some diversity exists among populations with regards to the morphology of the female cone and seeds. Other studies, with material from Poland, also reported similar results [18,50]. Differences in seed morphology are known to have an impact on seed dispersal in silver fir [51]. The morphometric variation of seeds and female cones has also been observed in other species, such as *Picea glauca* [52]. The variation we observed among populations may be the result of environmental and/or genetic differences, as well as their interaction. In this respect, adaptation to local conditions is a main reason for changes in morphology in forest species [53].

The germination capacity of silver fir seeds is usually low [18]. It has been previously shown that seed germination is positively correlated with seed weight, so that, for example, seeds weighing over 60 mg had a foreseen sprouting ability above 40% [46]. However, we did not find a significant relationship between both parameters, suggesting that, in the populations evaluated here, seed weight is not a major determinant of seed germination. Our results also indicate that germination capacity is greatly affected by the way seeds are stored before sowing. The highest germination capacity for all populations was observed using the storage method in wet sand previously described [32]. Significant differences were also observed in the germination capacity of the different silver fir populations studied, as can be observed in Table 4, suggesting different levels of fitness [51].

The 12 SSR markers [21,36,37] that were polymorphic in the *A. alba* materials included in the present work allowed us to evaluate the diversity and genetic structure of the nine Romanian populations under consideration, using three multiplexes. Some reports on silver fir have used the same, or some of the same, markers as those in our study [21,22,24–26,54]. The calculated diversity statistics per locus are generally lower than those obtained earlier for Italian or Romanian populations [21,22,24,54]. However, when compared to other studies in Serbian populations [25,26], the number of alleles in the shared markers is higher in our case. The values of observed heterozygosity (Ho) for 10 out of the 12 SSR markers evaluated are considerably lower than those of expected heterozygosity (He), indicating a higher proportion of homozygotes than expected in panmixis. This results in high values for the fixation index (Fis), which are in turn considerably higher than those found in Spanish populations of

*A. alba* [1]. The PIC values per locus are variable, but the average is moderate, even though some SSR markers could be considered as having a high PIC, with values above 0.5 [55]. The levels of diversity of genomic SSRs were on average higher than those of EST-SSRs, as found before in silver fir [21], and as is common in forest trees and in field and vegetable crops [56–58].

The average number of alleles per locus per population was similar to that observed in other Romanian populations [22] but lower than those found in a more recent study [54]. It should be considered, however, that the former study used EST-SSR markers, whereas the latter was based on gSRRs. The observed heterozygosities of the populations selected for the present work were generally lower than those of other former studies, including the two mentioned above [22,24,54]. Accordingly, the fixation indexes were quite high, suggesting that processes of consanguinity may be occurring in these populations, which, in the end, might lead to inbreeding depression [47].

The evaluation of the genetic distances between populations revealed a low relationship with geographical distances. For example, despite their relative geographical proximity, the five populations from the northwestern part of Romania did not display closer genetic distances between them than with those farther away. A similar phenomenon was reported in Romanian populations of *Larix decidua* [59,60]. Cluster analysis confirmed that the genetic relationships between populations were only marginally associated with geographical distances. In this respect, other authors [54] have also found that Romanian *A. alba* populations from different geographical areas of the Carpathians clustered in separate branches of a phylogenetic tree. However, within a geographical area, the relationship of geographical distance and the position in the dendrogram was weak.

AMOVA analysis shows that there is little differentiation between populations, with only 9% of the total variance attributable to population differences. These values are similar to those observed in Italian silver fir populations [24], and other authors [22,54] have also reported low levels of genetic differentiation for Romanian populations. It has been suggested [22] that Romanian silver fir populations probably have a common origin in the same glacial refugia in the central Balkans, which might have contributed to the low levels of genetic differentiation. These low values are common in conifers due to their mostly allogamous mating system [47] and have also been found in Romanian *L. decidua* [60].

The fact that we have found evidence of a positive correlation between seed germination and genetic diversity, but a negative one with the inbreeding coefficient, may have importance for the long term viability of *A. alba* populations, particularly in a climate change scenario [53], as inbreeding may result in reduced fitness and decline of forest populations [47]. Therefore, management of extant populations and reforestation programmes of *A. alba* should include sufficient genetic diversity to avoid consanguinity and inbreeding depression.

In conclusion, we have found significant variability in the morphological characters of female cones and seeds from the nine selected Romanian populations of silver fir. Variation existed between populations in a germination capacity, which may have an impact on their fitness. Moreover, germination also depended to a large extent on the seed storage procedure used before germination; in this respect, the wet sand storage method was consistently the best for enhancing germination of silver fir seeds. The nine sampled populations also displayed considerable genetic diversity and a low degree of genetic differentiation. The fact that, for some of the populations, we detected relatively high degrees of fixation might have implications for the long-term viability of these populations. Therefore, our results suggest that low levels of diversity and a high degree of inbreeding may reduce germination, reducing fitness and potentially compromising the populations' viability. Our results complement previous studies on Romanian populations of silver fir and provide relevant information for the conservation and management of genetic resources and reforestation programmes of *A. alba* in Romania.

**Author Contributions:** Conceptualization, J.P. and R.E.S.; methodology, I.M.T.M., M.B., A.F.S., M.P.; software, S.V. and J.P.; validation, S.V. and O.V.; formal analysis, I.M.T.M., M.P., A.F.S.; investigation, I.M.T.M., S.R., A.F.S. and M.P.; resources, J.P.; data curation, I.M.T.M., M.P., L.H. and A.F.S.; writing—original draft preparation, I.M.T.M. and M.P.; writing—review and editing, I.M.T.M., M.P., J.P.; visualization, S.V. and J.P.; supervision, O.V., J.P. and M.P.; project administration, R.E.S.; funding acquisition, J.P and M.P. All authors have read and agreed to the published version of the manuscript.

**Funding:** The publication was supported by funds from the National Research Development Projects to Finance Excellence (PFE)-37/2018-2020 granted by the Romanian Ministry of Research and Innovation. The first author (I.M.T.M.) is thankful to UPV (Valencia) and the Erasmus+ mobility programme, financed by the European Commission, for her scholarship in UPV. The research was partially supported by the Advanced Horticultural Research Institute of Transylvania (ICHAT), University of Agricultural Sciences and Veterinary Medicine of Cluj-Napoca (USAMVCN), and Polytechnic University of Valencia (UPV). Mariola Plazas is grateful to Generalitat Valenciana and Fondo Social Europeo for a post-doctoral grant (APOSTD/2018/014).

**Conflicts of Interest:** The authors declare no conflict of interest.

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
