# Peer review of "Genetic Relationships and Reproductive Traits of Romanian Populations of Silver Fir (Abies alba): Implications for the Sustainable Management of Local Populations"

_sustainability, doi:10.3390/su12104199_

Round 1

Reviewer 1 Report

Dear authors

Present manuscript entitled “Genetic relationships and reproductive traits of Romanian populations of silver fir (Abies alba):  Implications for the sustainable management of local populations” contained an impressive nine native populations from different geographical areas as samples and was very well planned. Manuscript is well written with adequate methods explained compiled results and corresponded well with the discussion. Present research may contain very useful information for conservation and management of genetic resources and reforestation programmes of A. alba. The results may be of high interest for conservation biologists and I hereby recommend the manuscript for consideration to publish in present form.

Wish you best

Thank you

Author Response

Reviewer’ comments

Dear authors

Present manuscript entitled “Genetic relationships and reproductive traits of Romanian populations of silver fir (Abies alba):  Implications for the sustainable management of local populations” contained an impressive nine native populations from different geographical areas as samples and was very well planned. Manuscript is well written with adequate methods explained compiled results and corresponded well with the discussion. Present research may contain very useful information for conservation and management of genetic resources and reforestation programmes of A. alba. The results may be of high interest for conservation biologists and I hereby recommend the manuscript for consideration to publish in present form.

RESPONSE: Thank you very much for your appreciation.

Reviewer 2 Report

The authors present a broad study of nine populations Abies alba in Romania. The investigations are miscellaneous and its scientific value is undoubted. The article is important in practical terms and especially at the current time when reforestation in European countries is relevant. Thus, the publication may extend the knowledge on conservation and management of A. alba genetic resources.

The title corresponds to the presented investigations, the keywords are adequate.

Abstract is adequate and contains short information on methods, results and main conclusions of this study.

Introduction. The state-of-the-art is covered in the introduction chapter and the relevant references are discussed. This chapter substantiate the aim and the significance of this study.

Materials and Methods. Research material and methods as well as statistical analyses are presented in the Materials and Methods section.

The authors should explain why the populations selected for the study are located at such distances, that is the populations 1,3,6, 7 and 8 are quite close to each other, and others such as 2 and 5 are very distant?

How many cones of each population were collected to determine length, diameter etc. The selection of adult trees was also not showed.

Results. The results of the study are discussed in detail and tables and figures are appropriate and visualize the research.

Discussion. In this section, the authors discuss their research findings in the context of other studies.

The conclusions are written properly and are focused on the issues raised in the study.

Author Response

Reviewer’ comments

The authors present a broad study of nine populations Abies alba in Romania. The investigations are miscellaneous and its scientific value is undoubted. The article is important in practical terms and especially at the current time when reforestation in European countries is relevant. Thus, the publication may extend the knowledge on conservation and management of A. alba genetic resources.

The title corresponds to the presented investigations, the keywords are adequate.

Abstract is adequate and contains short information on methods, results and main conclusions of this study.

Introduction. The state-of-the-art is covered in the introduction chapter and the relevant references are discussed. This chapter substantiate the aim and the significance of this study.

Materials and Methods. Research material and methods as well as statistical analyses are presented in the Materials and Methods section.

RESPONSE: Thank you very much for your appreciation.

The authors should explain why the populations selected for the study are located at such distances, that is the populations 1,3,6, 7 and 8 are quite close to each other, and others such as 2 and 5 are very distant?

RESPONSE: The majority of the population selected are included in the National Catalogue of Forest Reproductive Materials of Romania. Populations 8 and 9 were collected in the Apuseni Mountains (western Carpathians), also the area of origin of populations 1,3, 6 and 7, mountains with a good presence of fir. In order to assess the effect of geographical distance on populations differentiation we selected accessions from different ranges of geographical distances. In this way, we have added the following sentence in Material and Methods section 2.1 (Plant Material): “Different ranges of distances among populations were chosen to evaluate the effect of geographical distance on genetic distance.”

How many cones of each population were collected to determine length, diameter etc. The selection of adult trees was also not showed.

RESPONSE: For each population, a large number of cones were sampled in bulk collection from selected healthy trees with good qualities for propagation in local nurseries. Out of these stocks 36 cones for each population were randomly taken. In Material and Methods  section 2.2 (Characterization of Female Cones and Seeds, and Seed Germination) we have indicated that “Female cones of A. alba, collected in bulk from adult trees from the nine selected populations…” and “A total of 36 cones from each population, randomly taken from the bulk sampling were used for the measurements.”

Results. The results of the study are discussed in detail and tables and figures are appropriate and visualize the research.

Discussion. In this section, the authors discuss their research findings in the context of other studies.

The conclusions are written properly and are focused on the issues raised in the study.

RESPONSE: Thank you very much for your appreciation.

Reviewer 3 Report

This manuscript samples genetic and phenotypic data from nine populations of silver fir to assess the genetic and reproductive trait diversity. The information is valuable for management, propagation, and conservation of this species. The authors find interesting results dealing with the best storage method for seeds and evidence of inbreeding potentially impacting future population fitness. The manuscript was well written and I had very few comments overall. My only major concern was with how individual trees in populations were chosen for sampling and if there is a difference in the size of these individuals between populations that could influence the differences in the morphological traits between populations.

Line 49: Can you clarify what you mean by “highest tree species”. Maybe say “Silver Fir is also found at higher altitudes compared to most tree species of the Abies genus, usually between 500 and 2000 m, …”

Line 63: Is there any information about what percent of the forest silver fir made up in the past in Romania? If this is known, it would be beneficial to add it here to contrast with the fact that there is only 5% now. If this information is not known or the percent has been the same then this sentence is good as is.

Line 65: change “educational or…” to “educational and…”

Line 112: Were the mean values determined for populations? If so, can you include how many trees were samples per population.

Line 176: Would differences in tree size influence the morphological characteristics? For instance, if all the individuals sampled in population 7 were larger than in population 1, would you then expect population 7 to have a larger cone length? I am concerned that these values would be correlated with DBH.

Line 211: Change “Other” to “Another”

Line 255: I would change this sentence to say something along the lines of “despite a positive trend in the relationship between genetic distances and geographic distances the relationship was nonsignificant” since P = 0.0967. You can also make this change in the abstract on Line 35.

Line 295: In the discussion as a whole, it would benefit from referencing back to the particular results of this paper when comparing them to other work. For example, line 318-319, add (Table 4) to the end of the sentence so readers know exactly where to go review the particular data points of this result, if they so choose.

Line 323: change “markers than in…” to “markers as those in…”

Author Response

Reviewer’ comments

This manuscript samples genetic and phenotypic data from nine populations of silver fir to assess the genetic and reproductive trait diversity. The information is valuable for management, propagation, and conservation of this species. The authors find interesting results dealing with the best storage method for seeds and evidence of inbreeding potentially impacting future population fitness. The manuscript was well written and I had very few comments overall.

RESPONSE: Thank you very much for your appreciation.

My only major concern was with how individual trees in populations were chosen for sampling and if there is a difference in the size of these individuals between populations that could influence the differences in the morphological traits between populations.

RESPONSE: Only adult, healthy trees were considered, with similar size. In this respect, we have indicated in the text that female cones were “…collected in bulk from healthy adult trees from the nine selected populations,...”.

Line 49: Can you clarify what you mean by “highest tree species”. Maybe say “Silver Fir is also found at higher altitudes compared to most tree species of the Abies genus, usually between 500 and 2000 m, …”

RESPONSE: Thank you for the corrections. Silver fir is the tallest tree of the genus Abies in Europe. Under favourable conditions, the species can reach an age of 500 to 600 years and tree reaches up to 60 m. We have changed the sentence changing “highest” by “tallest”: “Silver fir is one of the tallest tree species of the Abies genus, under favorable conditions up to 60 m”.

Line 63: Is there any information about what percent of the forest silver fir made up in the past in Romania? If this is known, it would be beneficial to add it here to contrast with the fact that there is only 5% now. If this information is not known or the percent has been the same then this sentence is good as is.

RESPONSE: A new paragraph has been introduced at line 63. “If in the remote past the area occupied by fir probably represented 10-15% of the forest area of Romania, in 1929 its share was reduced to 6.5%, and in 1984 the tree occupied 5.12% of the Romanian forests. In 1989, it occupied about 317 thousand hectares, i.e. 5.1%, and in 2003 it occupied 307 thousand hectares, which means 5.0% of the area covered by forests [10]

Line 65: change “educational or…” to “educational and…”

RESPONSE:  Done.

Line 112: Were the mean values determined for populations? If so, can you include how many trees were samples per population.

RESPONSE: The values represent mean values per populations. The number of individual is specified on lines 133-134 (n = 5 for populations 1, 3, 8, and 9; n = 4 for populations 2, 4, 5, 6, and 7).

Line 176: Would differences in tree size influence the morphological characteristics? For instance, if all the individuals sampled in population 7 were larger than in population 1, would you then expect population 7 to have a larger cone length? I am concerned that these values would be correlated with DBH.

RESPONSE: No apparent differences in size were observed among trees of the different populations and we did not appreciate a relationship between variation inmorphological traits of cones and tree sizes, as samples were randomly taken from a bulk collection in each population.

Line 211: Change “Other” to “Another”

RESPONSE: Done.

Line 255: I would change this sentence to say something along the lines of “despite a positive trend in the relationship between genetic distances and geographic distances the relationship was nonsignificant” since P = 0.0967. You can also make this change in the abstract on Line 35.

RESPONSE: The sentence at line 255 was changed. It was also mentioned in the abstract at Line 35.

Line 295: In the discussion as a whole, it would benefit from referencing back to the particular results of this paper when comparing them to other work. For example, line 318-319, add (Table 4) to the end of the sentence so readers know exactly where to go review the particular data points of this result, if they so choose.

RESPONSE: Table 4 was mentioned at line 318.

Line 323: change “markers than in…” to “markers as those in…”

RESPONSE: Done.